# Appropriate Use of Antibiotic and Principles of Antimicrobial Stewardship in Children

**DOI:** 10.3390/children10040740

**Published:** 2023-04-17

**Authors:** Melodie O. Aricò, Enrico Valletta, Désirée Caselli

**Affiliations:** 1 U.O. Pediatria, Ospedale G.B. Morgagni—L. Pierantoni, AUSL Romagna, 47121 Forlì, Italy; melodieolivialoredanarosa.arico@auslromagna.it (M.O.A.); enrico.valletta@auslromagna.it (E.V.); 2 U.O.C. Malattie Infettive, Ospedale Pediatrico Giovanni XXIII, Azienda Ospedaliero-Universitaria Consorziale Policlinico di Bari, 70100 Bari, Italy

**Keywords:** stewardship, antibiotics, de-escalation, antimicrobial resistance

## Abstract

Antibiotics account for over 10% of the overall drug expense of the National Health System in Italy in 2021. Their use in children is of particular interest on one side, because acute infections are very common in children, while they build their immunologic library of competence; on the other side, although many acute infections are expected and turn out to be of viral origin, caregivers will often ask the family doctor or primary care attending to reassure them by prescribing antibiotic treatment, although it may often be unnecessary. The inappropriate prescription of antibiotics in children may likely be a source not only of undue economic burden for the public health system but also of increasing development of antimicrobial resistance (AMR). Based on those issues, the inappropriate use of antibiotics in children should be avoided to reduce the risks of unnecessary toxicity, increase in health costs, lifelong effects, and selection of resistant organisms causing undue deaths. Antimicrobial stewardship (AMS) describes a coherent set of actions that ensure an optimal use of antimicrobials to improve patient outcomes while limiting the risk of adverse events including AMR. The aim of this paper is to spread some concept of good use of antibiotics for pediatricians or every other physician involved in the choice to prescribe, or not, antibiotics in children. Several actions could be of help in this process, including the following: (1) identify patients with high probability of bacterial infection; (2) collect samples for culture study before starting antibiotic treatment if invasive bacterial infection is suspected; (3) select the appropriate antibiotic molecule based on local resistance and narrow spectrum for the suspected pathogen(s); avoid multi-antibiotic association; prescribe correct dosage; (4) choose the best route of administration (oral vs. parenteral) and the best schedule of administration for every prescription (i.e., multiple administration for beta lactam); (5) schedule clinical and laboratory re-evaluation with the aim to consider therapeutic de-escalation; (6) stop antibiotic administration as soon as possible, avoiding the application of “antibiotic course”.

## 1. Introduction

Antibiotics account for over 10% of the overall drug expense of the National Health System in Italy in 2021 [1]. Their importance is obvious inasmuch that they allowed, progressively over the last 80 years, to tackle bacterial infections, including some of the most frequent and lethal transmissible human diseases. The next question is: are antibiotics used appropriately? This is not only an economical issue but also an issue of safety as well as of continuous and maintained efficacy. In 2019, the last evaluable year of the pre-COVID-19 era, the rate of prescription of antibiotics in a country with affluent economic resources such as Italy was of 822 prescriptions/1000 children per year. This prescription rate dropped by 50% during the following 2020 year, during the lock-down season [2]. This may suggest that the use of antibiotics by pediatricians is influenced also by “cultural” attitudes.

During the first decades following their discovery, antibiotics were regarded as an everlasting weapon, which favored their overuse. In the meanwhile, some microorganisms were on their way to progressively develop tools and skills to evade the effect of antibiotics, thus jeopardizing our ability to treat infectious diseases expected to be potentially or likely curable. As a result, the worldwide spread of the use of antibiotics increased the selective pressure on the global microbial population, thus inducing a significant increase in antimicrobial resistance (AMR) [3].

Among other warnings on antibiotic overuse in pediatric patients, who have a longer follow up, there are also reported increases in the prevalence of asthma [4], allergic disease (atopic eczema, food allergy) [5], and risk of childhood obesity and inflammation [6].

Based on the above, pediatricians should keep in mind that the use of antibiotics should always be driven by precise principles and information. Antimicrobial stewardship (AMS) describes a coherent set of actions that ensure an optimal use of antimicrobials to improve patient outcomes while limiting the risk of adverse events including AMR. The introduction of AMS programs in hospitals is part of most national action plans to mitigate AMR, yet the optimal components and actions of such a program remain undetermined.

## 2. What about Antimicrobial Resistance?

AMR happens when germs such as bacteria, virus and fungi develop the ability to defeat the drugs designed to kill them. AMR is a naturally occurring process. However, any increase in AMR is driven by a combination of germs exposed to antibiotics and the spread of their resistance mechanisms. Mechanisms may be different, from modifying the drug target to eliminating drugs from the microbial cell before they can reach their target [7,8]. The final effect of AMR will be a loss of drug effectiveness, the need for different molecules to treat infections and search for new drugs, prolonged hospitalizations and therapies, and an increase in mortality [9,10].

Widespread exposure to antibiotics eliminates susceptible strains and selects resistant ones. Selective pressure will act without distinction both on pathogens that were the intended target of antibiotic therapy, but also on other microorganisms that colonize the host (animal or human being), and natural elements (water, soil, etc.) [11]. Therapeutic antibiotic administration also impacts on microbiome, killing “helpful germ” and allowing resistant microorganism to become prevalent [12]. Resistant strains will spread their resistance both by replication (vertical transmission) and by transferring their competence to different germs, thus resulting in new resistant strains (horizontal transmission) [8].

As a result of AMR spread, any patient who received antibiotics could afterwards be affected by resistant infection, but the local and global diffusion of resistant germs is not less important for the community. In fact, the inappropriate use of antibiotics (as listed above) is a driving factor to exert selective pressure on microbial populations, which results in resistant strains selection [13,14]: indeed, the main mechanisms sustaining AMR are an inappropriate use of antimicrobials, overuse, incorrect dosing, and extended duration.

On the basis of the statistical models used by “Antimicrobial Resistance Collaborators”, there were an estimated 4.95 million deaths associated with bacterial AMR in 2019, including 1.27 million deaths attributable to bacterial AMR [10].

Another important aspect is the economic effect, which includes direct and indirect cost. The direct cost of illness includes cost of hospitalization and medication. The indirect cost of illness comprises the present and future costs to society from morbidity, disability, premature death, and in particular the reduced effective labor supply (due to the lower productivity and deaths of workers) [15].

The World Bank Group published a report presenting different scenarios, estimating the effect of AMR on the global economy: in the optimistic scenario, with the lower effect of AMR, the global economic output is projected to be 1.0% lower by 2030 and 1.1% lower by 2050 than in the standard World Bank long-term projection for the global economy that excludes AMR [15].

The global action plan on antimicrobial resistance by the World Health Organization (WHO) [9] outlines five objectives: some are of global or national level, such as “to develop the economic case for sustainable investment that takes account of the needs of all countries and to increase investment in new medicines, diagnostic tools, vaccines, and other interventions” or “to strengthen the knowledge and evidence base through surveillance and research”. Other objectives are of local, institutional and personal level, such as “to reduce the incidence of infection through effective sanitation, hygiene and infection prevention measures”; “to improve awareness and understanding of antimicrobial resistance through effective communication, education and training” and “to optimize the use of antimicrobial medicines in human and animal health”.

On one side, health care providers’ perception and attitude regarding AMR is often disregarded [16]; otherwise, physicians may run the risk of remaining focused on single human (i.e., their current patient) infectious disease, which may be pretty good for that patient, but it is myopic if this prevents the prescribing specialist from looking at the landscape of the impact of antibiotic prescriptions, including his/her one, on global health. Thus, improving health care providers’ awareness of AMR importance has a great value to optimize antimicrobic, especially antibiotic, prescription in hospital and local settings, which in turn will progressively affect a much wider landscape.

The aim of this paper is to focus the non-specialist reader’s attention on evaluations and behaviors which build up, in a single word, a more appropriate use of antibiotics.

### Antimicrobial Stewardship

Antimicrobial stewardship (ABS) involves optimizing antibiotic use while using cost-effective interventions to minimize antibiotic resistance [17]. The ABS is the set of indications and good practice to optimize the use of antibiotics—in terms of molecule choice and treatment duration—aiming at the best clinical result with the lower risk of adverse events and of inducing AMR [18,19,20].

Generally, ABS programs include a multidisciplinary institutional team that organizes surveillance on antimicrobial prescription, optimizes prescription process (i.e., antibiotic restriction), creates regular infectious diseases team consultation, and takes care of educational programs on appropriate antibiotic use. ABS programs have been more often run in adult patient settings. The specificity of pediatric patients, made of lower total numbers, but also their spreading from primary care to the ICU [21,22,23], can make starting an ABS pediatric program more difficult than in the adult population [24]. Building an ABS pediatric program means also creating a multidisciplinary interprofessional ABS team able to prepare guidelines, monitoring antibiotic prescriptions and defining corrective measures, and the availability of administrative consensus with adequate financial support [25].

Overall, the major goal of an ABS program is the education of pediatricians so that awareness of the antimicrobial stewardship’s principles are shared with physicians in the hospital and outside it, so that all of them may become proactive in refining an antimicrobial prescription.

## 3. Good Use of Antibiotics in Children

### 3.1. Patient Selection

The first step is to avoid unnecessary treatments. Some simple questions may help physicians: is antibiotic therapy mandatory for that patient, at that time and for his clinical condition? Might it be a viral infection? May a wait-and-re-evaluate approach be appropriate for that patient before starting treatment?

In a patient in good condition, with a short-lasting history of mild symptom, a wait and re-evaluate (after 24–48 h) approach is widely acceptable. Spontaneous improvement often occurs suggesting the benign course of the infection, especially for upper respiratory tract infections. Re-evaluation and delay to start antibiotic therapy is often a useful strategy to avoid unnecessary treatments. Caregivers may play a key role in deciding to delay prescription: if caregivers are attentive and receptive, delay can be easier and safe, sharing with parents the opportunity for clinical re-evaluation in case of worsening. Sometimes, parents/caregivers may be non-compliant for a wait-and-see strategy: they will claim an immediate start of antibiotic therapy, thus inducing the physician to prescribe an antibiotic therapy which appears likely unnecessary, given the likely viral origin of the infection. Caregivers’ education will be pivotal: it is much better if they are already informed on the difference between viral and bacterial infections, the lack of advantage, and the risk of disadvantage (adverse reaction, side effect, AMR) if antibiotic prescription is inappropriate. The time spent to educate patients and caregivers will result in saved antibiotic doses, avoidance of side effects and reduced risk of AMR.

Prescribing antibiotics in patients with suspected or proved viral infection in a good clinical condition, with the aim of preventing bacterial over-infection, should always be avoided, since it would have the sole effect of modifying the commensal flora [13,14] and letting the risk of AMR raise.

Bacterial infection may certainly occur as a complication of surgery. However, is “preventive” administration, i.e., antibiotic prophylaxis, useful for this patient? It may be certainly useful but only in very limited situations: clean (e.g., implant prosthesis, cardiac surgery for malformations) or clean contaminated surgery. In fact, consequences of bacterial infection in this surgery can lead to life-threatening complications. Neutropenic patients may deserve prophylactic antibiotic, while the role of prophylaxis in dental maneuvers in cardiopathic patients or in malformations of the urinary tract has been greatly reappraised.

### 3.2. Sample Collection for Cultures and Baseline Evaluation

Bacterial isolates from culture of blood, urine, cerebrospinal fluid or secretions are of the greatest importance to identify the pathogen, explore its pattern of sensitivity/resistance and thus optimize the antibiotic prescription for a possible or documented invasive bacterial infection [26].

Especially in hospital settings, physicians should collect laboratory information (e.g., basal inflammatory biomarkers) likely of interest to modulate antibiotic therapy over the coming days and should not forget to accumulate data on bacterial isolates to build local data on bacterial epidemiology.

### 3.3. Which Antibiotic Should Be Prescribed?

Once biologic samples have been collected and basal evaluation is complete, antibiotic therapy can start. To choose initial, empiric antibiotic therapy, some specific issues may be of support:-Previous microbiological isolates from the same patient, if available;-Local data on antimicrobial resistance, whether information is available;-To choose the antibiotic with the narrowest antimicrobial spectrum (the range of microorganisms an antimicrobic agent can kill or inhibit), related to the patient clinical condition;-Avoid multi-agent prescription, i.e., two or more molecules at the same time; when multiple therapies are necessary, avoid possible overlapping of antibacterial spectra (antibiotics effective against the same microbial agents);-Prescribe the appropriate dosage, paying attention that the antibiotic is not under dosed. Administration of suboptimal dosage, with the aim of reducing the risk of side effects, is not a good choice, since what is really reduced is only the probability to be effective. For example, prescription of amoxicillin is often underdosed, with dosage inferior to 60 mg/kg/die [27,28,29];-Choose the most appropriate way of administration: parenteral administration should be the first choice in septic patients, especially if with hemodynamic instability, or in case of difficult oral administration (vomiting, gastrointestinal dysfunction, or unconsciousness). In every other situation, oral administration could be the first choice given its non-inferiority in many infectious diseases, such as hospitalized children with severe not complicated pneumonia [27,30], bone and joint infection [31,32,33], acute pyelonephritis in children older than 3 months old, without urinary tract abnormalities [34,35].

Furthermore, intravenous administration should be avoided if not strictly necessary because of the non-molecule-specific risk of adverse effects such as phlebitis, local and vascular infection, excess fluid administration and patient discomfort [36,37,38,39,40].

#### 3.3.1. Local Information on Microbial Resistances

National and international surveillance on antimicrobial resistance have been developed in the last few decades. About 85% of countries/areas in the WHO European Region have an AMR national/area action plan. The European Centre for Disease Prevention and Control (ECDC) and the WHO Regional Office for Europe published a joint report on AMR data from invasive isolates in Europe [41]. The report shows that AMR is widespread in the WHO European Region: a north-to-south and west-to-east gradient was generally observed, with higher AMR percentages in the southern and eastern countries of Europe, although the AMR situation varied depending on the bacterial species, antimicrobial group, and geographical region. AMR percentages for the bacterial species–antimicrobial group combinations under surveillance continue to be high with carbapenem resistance in *Escherichia coli* and *Klebsiella pneumonia* and vancomycin resistance in *Enterococcus faecium* increased during 2016–2020. High rates of resistance to third-generation cephalosporins and carbapenems in *Klebsiella pneumoniae*, of carbapenem-resistant *Acinetobacter* species and *Pseudomonas aeruginosa* in several countries in the European Region are of concern [41]. 

An up-to-date knowledge of local AMR data is one of the most important resources to limit the use of broad-spectrum antimicrobials, especially when a narrower spectrum molecule has comparable effectiveness [41,42].

#### 3.3.2. AWaRe Classification

The World Health Organization (WHO) has implemented, in the context of a global project to combat AMR (Global Action Plan on Antimicrobial Resistance) [9], a classification system for antibiotics called AWaRe (Access, Watch, Reserve), which divides the most used antibiotics (currently 257 molecules) into three groups based on the potential to induce AMR rather than to classify their antimicrobial effectiveness, [Table 1] thus aiming to monitor their use:The Access group includes first and second-choice antibiotics to treat the most common conditions with the best therapeutic value, minimizing the potential to induce resistance. Frequently for these antibiotics, oral administration is possible (Figure 1).The Watch group includes a selection of antibiotic molecules that should be used for specific and limited infections either as a first or second-choice agent. Their prescription is generally the target of antimicrobial stewardship programs for their higher potential to induce AMR.The Reserve group should be regarded as the final resource against multi-resistant bacteria when all previous antibiotic treatments have already failed. They are intended only for hospital use, and their prescription is usually under strict control, single-patient oriented, and better with specialist consultation.

#### 3.3.3. Schedule of Antibiotic Administration

Some basic knowledge of pharmacokinetic characteristics of different classes of antibiotics is important to decide the best schedule of administration (dosage and number of daily doses) of an antibiotic.

Beta-lactam antibiotics (penicillins, cephalosporins, carbapenems, monobactams) are frequently prescribed in children. The pharmacokinetic/pharmacodynamic (PK/PD) target for antimicrobial success is the fraction of time that the unbound drug concentration at the infection site is above the MIC (T > MIC) for every specific pathogen [28].

An optimal administration of time-dependent antibiotics (i.e., beta-lactam, vancomycin, clindamycin) should try to reach a stable plasma concentration above the MIC for all times of antibiotic treatment. If intravenous administration is started, the ideal schedule, currently under evaluation for many parenteral molecules (piperacillin–tazobactam, vancomycin, etc.) provides a loading dose to reach the effective concentration above the MIC, which is followed by continuous infusion to ensure T > MIC is maintained as long as possible [44,45] (Figure 2). If antibiotics are administered orally, multiple daily doses should be preferred (Figure 3): i.e., for amoxicillin, administration every 8 h is better than every 12 h [29].

On the other hand, the effectiveness of concentration-dependent drugs is determined by the maximum concentration reached in a single point of the kinetic curve. The reference variables are the maximum concentration (Cmax) that is reached beyond the MIC (Cmax/MIC) and the area under the curve (AUC) that is above the MIC (AUC/MIC). The higher the values reached by these two variables, the greater the antibacterial effect (Figure 4).

### 3.4. Clinical and Laboratory Re-Evaluation: Therapeutic De-Escalation

The antibiotic administration must be questioned daily to confirm the need to prolong treatment.

The concept of “antibiotic course”—i.e., administration for a predetermined time, to be completed regardless of patient clinical response and improvement—is obsolete [46,47]. Continuing an antibiotic treatment that is no longer necessary can induce the selection of AMR without any further clinical benefit [13,14]. Therefore, the duration of therapy should not be established a priori but rather considered, day-by-day, based on new clinical and laboratory information that may become available.

#### 3.4.1. Early Switch to Oral Administration

Early switch from parenteral to oral administration is recommended—even after 48 h—when the patients show a significant clinical improvement (afebrile, hemodynamically stable, in good general conditions) and the inflammation markers (generally C-reactive protein) show a favorable trend. An early switch should be recommended toward a non-inferior oral treatment, also to reduce adverse effect, such as phlebitis, local and vascular infection, excess fluid administration and patient discomfort [36]. Switch to oral therapy may also be a step toward the early discharge of hospitalized patients.

#### 3.4.2. Re-Evaluation Based on Microbiologic Isolations

The re-evaluation of antibiotic treatment is mandatory as soon as an antibiogram is available: checking the susceptibility of the isolated microorganism to currently administered antibiotic(s) is necessary but not sufficient. The antibiotic administered should have the narrowest spectrum available for the isolated pathogen. Thus, when microbiological data become available, antibiotic treatment should be modified with a target therapy, ideally with a molecule active only against the isolated microorganism.

The use of broad-spectrum molecules should be limited to ensure patient safety: an inappropriately extended use of broad-spectrum antibiotics results in interference with commensal microorganisms and the selection of resistant strains. These microorganisms could then be responsible for severe and difficult to treat infections (e.g., Clostridium difficile).

Therefore, the concept of therapeutic de-escalation could be summarized as follows: as soon as the patient has clinically improved and/or new microbiological information becomes available, antibiotic treatment must be re-evaluated and possibly modified to use the minimum number of antibiotics, possibly just one, with the narrowest spectrum and consider parenteral to oral switch.

### 3.5. When to Stop Antimicrobial Treatment?

Many clinical trials have been conducted, and many are underway with the aim to define the standard antibiotic treatment duration for different infectious diseases, including urinary tract infections [48,49], pneumonia [50,51,52], neonatal Gram-negative bacterial sepsis [53] and septic arthritis [54].

Several studies confirm that patients well-appearing with uncomplicated pneumonia should be treated for no longer than 5 days [51], although the efficacy of even a 3-day treatment is under evaluation [50,55,56]. As for pneumonia, other infectious diseases could be treated with shorter duration of treatment. The decision to prolong antimicrobial treatment to complete an “antibiotic course” [46], even in patients with complete clinical recovery, may have the undesired effect to induce new AMR [13,56] without any clinical advantage. Thus, in every infectious disease, a reassessment of the real need of an extended treatment is important, since the discontinuation of antibiotic therapy, when safely feasible, will contribute to preventing AMR from increasing and to preserving the antibiotics effectiveness.

## 4. Conclusions

Antimicrobial prescriptions are frequent for children during the first years of life because of the frequent occurrence of febrile episodes. Yet, special attention should be paid also according to the implication that such prescription may have not only for the individual child (also in a life-long perspective) but also for the community. An excessive use of antibiotics facilitates an increase in AMR. Based on the above, starting an antibiotic therapy, apparently a frequent and low-intensity action, should instead be regarded as an occasion for the physician to implement quality in the daily work. Many critical issues can lead to a non-optimal use of antibiotics. An assessment list for prescribing antibiotics [Table 2] should keep in mind the most common mistakes; we propose the following:Inappropriate or “too early” start of antibiotic therapy even in a clinical picture suggesting a viral infection;Post-operative prophylaxis not indicated or continued beyond 24 h.Lack of microbiological analysis supporting antibiotic treatment;Non-optimal antibiotic selection: starting with a broad-spectrum antibiotic when another with narrower spectrum (i.e., amoxicillin—clavulanic acid or ceftriaxone vs. amoxicillin) could be used;Suboptimal dosage and administration schedule for the considered infectious disease; parenteral administration even if oral route is feasible and equally effective;No therapeutic de-escalation, i.e., no modification of broad-spectrum therapy with a narrow-spectrum one;Duration of antibiotic therapy based on a pre-established “antibiotic course”.

In conclusion, antibiotic prescription is a complex medical act, which requires a careful and thoughtful evaluation. Every physician should choose the best antibiotic for “that” patient, best route of administration, and time when antibiotic therapy can be discontinued; the initial decision should be reassessed upon the availability of new clinical or microbiological information. All these elements must be considered not only for the well-being of the single patient but also for the well-being of future individuals who might not benefit any more from a resource that, until now, has been viewed as granted and unlimited.

## Figures and Tables

**Figure 1 children-10-00740-f001:**
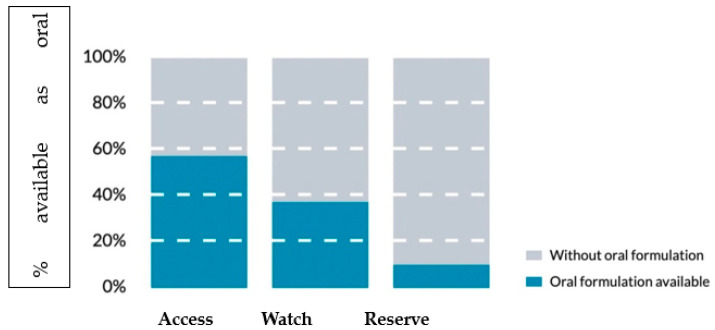
Percentage of oral formulations by AWaRe category (modified by [43]).

**Figure 2 children-10-00740-f002:**
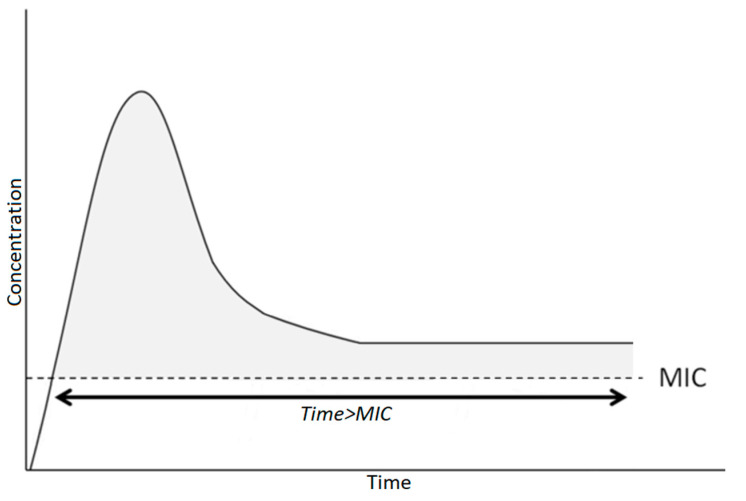
Pharmacokinetics of time-dependent drugs administered in continuous infusion.

**Figure 3 children-10-00740-f003:**
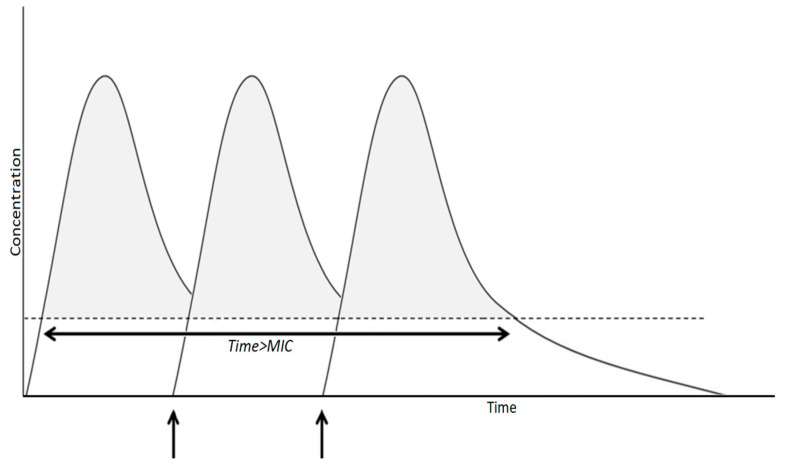
Time-dependent antimicrobial pharmacokinetic in intermittent administration. Antimicrobial efficacy is determined by the time above MIC (T > MIC) of plasmatic unbound drug concentration. Arrows shows the time of administrations.

**Figure 4 children-10-00740-f004:**
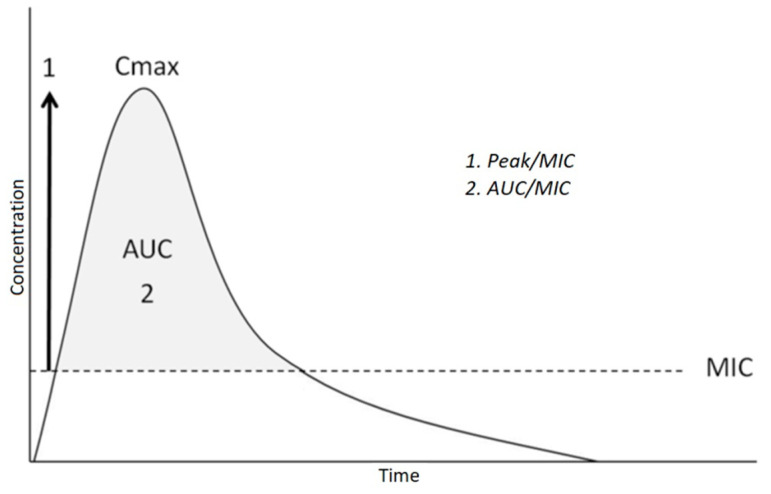
Kinetics of concentration-dependent drugs. The effectiveness of the drug is linked to how high the absolute peak (Cmax) and the area under the curve is above the MIC (AUC/MIC).

**Table 1 children-10-00740-t001:** AWaRe classification (modified by [43]).

**Group**	**Features**	**Example**
Access	First or second-choice antibiotics Offer the best therapeutic value while minimizing the potential for resistance	Amoxicillin, Amoxicillin–clavulanic acid, Cefalexin, Cefazolin, Clindamycin, Cotrimoxazole, Gentamicin;
Watch	First or second-choice antibiotics Only indicated for specific, limited number of infective syndromes more prone to be a target of antibiotic resistance and thus prioritized as targets of stewardship programs and monitoring	Azithromycin, Cefixime, Ceftriaxone, Vancomycin, Piperacillin–tazobactam, Meropenem
Reserve group	“Last resort” Highly selected patients (life-threatening infections due to multi-drug resistant bacteria) Closely monitored and prioritized as targets of stewardship programs to ensure their continued effectiveness	Ceftazidime–avibactam, Colistin, Daptomycin, Linezolid

**Table 2 children-10-00740-t002:** Assessment list for prescribing antibiotics.

**Step**	**Key Point**	**Question**
1. Patient selection	Bacterial infection probability	Is antibiotic treatment mandatory for this patient?
	Clinical re-evaluation after 24–48 h	
	Caregivers’ compliance	
	Prophylaxis	Clean or clean-contaminated surgery? High-risk patient?
2. Baseline evaluation	Suspect of invasive bacterial infection	Collect microbiological sample (i.e., blood, urine, swab)
	Laboratory test	Inflammatory index (WBC, CRP, ESR, PCT)
3. Choose the best antibiotic	Local data on AMR	
	Consider antibiotic with a narrow spectrum	
	Avoid combined therapy with more than a single antibiotic molecule, whenever possible	If using more than a single antibiotic is necessary, spectra overlapping should be checked
	Appropriate dosage (neither over- nor under-dosed) and schedule of administration	Dosage Number of daily doses
	Evaluate if oral administration is possible	Not indicated in sepsis or oral intolerance
4. Therapeutic de-escalation	Early switch from parenteral to oral route	Switch to oral administration when patient shows clinical and laboratory improvement
	Optimization of antibiotic treatment upon new microbiological information	Evaluate microorganism susceptibility to antibiotic treatment Consider a narrow spectrum antibiotic Consider to stop multiple antibiotic administration
5. Stop antibiotic treatment	Avoid prolonging antibiotic treatment to complete “antibiotic course”	Consider treatment interruption if clinical and laboratory resolution

## Data Availability

Data sharing not applicable.

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
