# Peer review of "Appropriate Use of Antibiotic and Principles of Antimicrobial Stewardship in Children"

_children, 2023, doi:10.3390/children10040740_

Round 1

Reviewer 1 Report

This review presents the overview of the 'ANTIMICROBIAL STEWARDSHIP’ in general but deviates from the main focus, ‘ANTIMICROBIAL STEWARDSHIP IN PEDIATRICS. I would like to request authors to revise the manuscript addressing following comments before considering this manuscript for publication in this journal.

1.     It would be appropriate if authors present clinical data on consequences of antibiotic overuse in children.

2.     Please present some examples of implementation and impact of pediatric antimicrobial stewardship in children.

3.     Please discuss the impact of antibiotic stewardship interventions regarding pediatric antimicrobial stewardship.

4.    Is there any difference between antimicrobial stewardship programs for children and adults?

5.     Please also discuss the limitations of antimicrobial stewardship programs for children, if any.

Author Response

This review presents the overview of the 'ANTIMICROBIAL STEWARDSHIP’ in general but deviates from the main focus, ‘ANTIMICROBIAL STEWARDSHIP IN PEDIATRICS’. I would like to request authors to revise the manuscript addressing following comments before considering this manuscript for publication in this journal. 1. It would be appropriate if authors present clinical data on consequences of antibiotic overuse in children. 2. Please present some examples of implementation and impact of pediatric antimicrobial stewardship in children. 3. Please discuss the impact of antibiotic stewardship interventions regarding pediatric antimicrobial stewardship. 4. Is there any difference between antimicrobial stewardship programs for children and adults? 5. Please also discuss the limitations of antimicrobial stewardship programs for children, if any.

Reviewer 2 Report

In this article nothing is new. All the information are already available in website and different guidelines provided by FAO. In addition, Including Abstract, Introduction all parts are not well-organized and also insufficient information.

Author Response

In this article nothing is new. All the information are already available in website and different guidelines provided by FAO. In addition, Including Abstract, Introduction all parts are not well-organized and also insufficient information.

Reviewer 3 Report

The manuscript, Antimicrobial Stewardship in Pediatrics, describes a checklist of recommendations for administering antibiotics as means of preventing antimicrobial resistance (AMR). The recommendations offer a list of things to avoid (such as the use of broad spectrum antibiotics and prophylaxis) as well as best practices for clinicians (such as collection of bodily fluids to assess biomarkers and consideration of pharmacokinetics when prescribing dosage)

Overall, I feel that the manuscript can be improved. The writing style seems to lack depth and nuance. Some statements are very simplistic and should be reconsidered. Some recommendations of assessment, such as giving the appropriate dose (line 97) or that antibiotics are an “important resource” (line 15) seem very obvious. The authors could provide citations for evidence of antibiotics overdosing to substantiate statements like this. 

Some statements can be combined into something more concise. For example, lines 25-28 could easily be merged into one statement.

The abstract is short. I suggest the authors state that they have developed an assessment list for prescribing antibiotics and enumerate each of the suggestions they have made.  

Lines 16, 56, 59, etc: avoid using pronouns such as “us” and “we”. It is convention to write articles in the third person.

There are numerous examples of terms being used that either oversimplify a concept or cause confusion. For example, “diners” in line 35. Instead of using an analogy to describe collateral selection it would be much clearer to offer a brief technical definition of it.

The “Steps of Stewardship” are very general and lack much detail.

Citations are missing after numerous statements.

Cite articles for mechanisms of drug resistance. (lines 29-30)

Likewise, it feels that there should be a citation after line 32. You are listing several significant effects of AMR and these must have some basis presented in the literature.

Line 44 are these global deaths?

Line 45 What monetary currency is this figure based on? Is there a time frame for the cost expenditures in references 5 and 6?  It isn’t clear to me what cost comparison is being made. 

Line 82 Omit “obviously”. If something is truly obvious it doesn’t need to be stated. 

Line 88 does empirical refer to trial-and-error in real-time? Having data from lab tests could likewise be considered empirical.

Line 94-96 clarify what is meant by “spectrum”. I’m assuming it refers to the range of targets that a drug exerts its effects. This should be clarified. 

Line 99 is there reasoning behind preference for oral administration? You bring this up several times but never state why oral is preferred.

Line 103-107 needs citation (especially since you mention a report)

Lines 113 -118 use italics for microbe names. Provide citations for these statements 

Fig 1 Provide a label for the y-axis (% available as oral formulation)

141-143 Access is not capitalized in the caption while watch and reserve are. Be consistent. Unless these are acronyms I suggest not using capitalization. Also, the capitalization of Aware in line 141 is inconsistent with its previous use in the text (AWaRe)

Fig 2 what do the arrows pointing along the x-axis indicate?

Cmax should include subscript for max 

Line 203 why is oral better?

Author Response

The manuscript, Antimicrobial Stewardship in Pediatrics, describes a checklist of recommendations for administering antibiotics as means of preventing antimicrobial resistance (AMR). The recommendations offer a list of things to avoid (such as the use of broad spectrum antibiotics and prophylaxis) as well as best practices for clinicians (such as collection of bodily fluids to assess biomarkers and consideration of pharmacokinetics when prescribing dosage) Overall, I feel that the manuscript can be improved. The writing style seems to lack depth and nuance. Some statements are very simplistic and should be reconsidered. Some recommendations of assessment, such as giving the appropriate dose (line 97) or that antibiotics are an “important resource” (line 15) seem very obvious. The authors could provide citations for evidence of antibiotics overdosing to substantiate statements like this. Some statements can be combined into something more concise. For example, lines 25-28 could easily be merged into one statement. The abstract is short. I suggest the authors state that they have developed an assessment list for prescribing antibiotics and enumerate each of the suggestions they have made. Lines 16, 56, 59, etc: avoid using pronouns such as “us” and “we”. It is convention to write articles in the third person. There are numerous examples of terms being used that either oversimplify a concept or cause confusion. For example, “diners” in line 35. Instead of using an analogy to describe collateral selection it would be much clearer to offer a brief technical definition of it. The “Steps of Stewardship” are very general and lack much detail. Citations are missing after numerous statements. Cite articles for mechanisms of drug resistance. (lines 29-30) Likewise, it feels that there should be a citation after line 32. You are listing several significant effects of AMR and these must have some basis presented in the literature. Line 44 are these global deaths? Line 45 What monetary currency is this figure based on? Is there a time frame for the cost expenditures in references 5 and 6? It isn’t clear to me what cost comparison is being made. Line 82 Omit “obviously”. If something is truly obvious it doesn’t need to be stated. Line 88 does empirical refer to trial-and-error in real-time? Having data from lab tests could likewise be considered empirical. Line 94-96 clarify what is meant by “spectrum”. I’m assuming it refers to the range of targets that a drug exerts its effects. This should be clarified. Line 99 is there reasoning behind preference for oral administration? You bring this up several times but never state why oral is preferred. Line 103-107 needs citation (especially since you mention a report) Lines 113 -118 use italics for microbe names. Provide citations for these statements Fig 1 Provide a label for the y-axis (% available as oral formulation) 141-143 Access is not capitalized in the caption while watch and reserve are. Be consistent. Unless these are acronyms I suggest not using capitalization. Also, the capitalization of Aware in line 141 is inconsistent with its previous use in the text (AWaRe) Fig 2 what do the arrows pointing along the x-axis indicate? Cmax should include subscript for max Line 203 why is oral better?

Round 2

Reviewer 2 Report

N/A

Reviewer 3 Report

The authors have addressed my previous comments and have improved the manuscript to be satisfactory.